# Genome Analysis of Anti-Phage Defense Systems and Defense Islands in *Stenotrophomonas maltophilia*: Preservation and Variability

**DOI:** 10.3390/v16121903

**Published:** 2024-12-10

**Authors:** Ghadeer Jdeed, Vera V. Morozova, Nina V. Tikunova

**Affiliations:** Institute of Chemical Biology and Fundamental Medicine, Siberian Branch of Russian Academy of Sciences, Prosp. Lavrentieva 8, Novosibirsk 630090, Russia; morozova@niboch.nsc.ru

**Keywords:** anti-phage defense systems, *S. maltophilia*, defense islands, phage therapy

## Abstract

Anti-phage defense systems are widespread in bacteria due to the latter continuous adaptation to infection by bacteriophages (phages). *Stenotrophomonas maltophilia* has a high degree of intrinsic antibiotic resistance, which makes phage therapy relevant for the treatment of infections caused by this species. Studying the array of anti-phage defense systems that could be found in *S. maltophilia* helps in better adapting the phages to the systems present in the pathogenic bacteria. Pangenome analysis of the available *S. maltophilia* strains with complete genomes that were downloaded from GenBank, including five local genomes, indicated a wide set of 72 defense systems and subsystems that varied between the strains. Seven of these systems were present in more than 20% of the studied genomes and the proteins encoded by the systems were variable in most of the cases. A total of 27 defense islands were revealed where defense systems were found; however, more than 60% of the instances of systems were found in four defense islands. Several elements linked to the transfer of these systems were found. No obvious associations between the pattern of distribution of the anti-phage defense systems of *S. maltophilia* and the phylogenetic features or the isolation site were found.

## 1. Introduction

*Stenotrophomonas maltophilia* is a gram-negative, rod-shaped, obligate aerobe and motile bacterium [1]. It is widespread and exhibits intrinsic high-level resistance to a variety of antibiotics [1,2,3]. Although *S. maltophilia* is a low-grade pathogen, it carries a large number of putative virulence factors [4] and the World Health Organization lists it as one of the leading drug-resistant pathogens in hospitals worldwide [2,5]. *S. maltophilia* strains display high phenotypic and genetic heterogeneity, forming what is called *S. maltophilia* complex [6]. This heterogeneity was confirmed by a number of studies at a phenotypic [7,8] and genotypic levels [9,10,11].

Due to the rise of antibiotic resistance in bacteria, bacteriophages (phages) are becoming a promising antimicrobials [12]. Most phages are specific to their hosts and, unlike antibiotics, they are able to adapt to the bacteria they co-exist with [13]. In turn, bacteria acquired various systems to protect themselves from the infecting phages over millions of years [14]. Some of these defense systems are highly conserved among different strains and species, while others are modified in a phage-dependent manner [15]. Examples of known bacterial anti-phage defense systems include: CRISPR-CAS, restriction modification (RM), restriction modification-like systems, systems that rely on phage protein sensing and sequence-independent detection of foreign nucleotide acids, abortive infection systems, toxin-antitoxin systems, and others [16].

It has been observed that anti-phage defense systems often tend to group in similar locations designated as a defense core, across different strains of the same bacterial species [9,10,11]. This allowed for discovering other systems in these locations in other bacterial genomes [17]. Pangenome analysis established anti-phage defense systems and their conserved occurrence sites for *Escherichia coli* [17] and *Pseudomonas aeruginosa* [18]. The parameters set for defining what constitutes a defense island are similar in these studies. In one study of anti-phage defense systems found in *E. coli*, a defense system-carrying mobile island was defined as a location in bacterial genome, where an anti-phage defense system was found in one strain. In addition, the island must contain more than ten genes including the anti-phage defense system, it should be flanked by five core genes (shared among more than 80% of *E. coli* strains), and the defense island should be empty in at least one strain [17]. According to another study of systems in *P. aeruginosa*, the defense core is flanked by specific marker genes, without necessary having instances of the island empty in some strains or including elements that would allow the defense islands to transfer between *P. aeruginosa* strains [18].

A comprehensive pangenome analysis of the anti-phage systems repertoire and their defense islands in the *S. maltophilia* genome has not been carried out. Such an analysis is relevant for phage therapy, as it sheds light on possible factors that can reduce or even eliminate the antibacterial activity of the phage toward the target pathogenic strain [19]. In this study, a pangenome analysis of 72 complete *S. maltophilia* genomes available in NCBI GenBank was performed, including five recently sequenced *S. maltophilia* strains that were isolated in Novosibirsk, Western Siberia. This analysis indicated the *S. maltophilia* anti-phage defense systems, their distribution and conservation, and defense islands in which they are likely to be found.

## 2. Materials and Methods

### 2.1. Bacterial Genome Sequence Analysis

*S. maltophilia* CEMTC 2142, 2355, 3659, 3664 and 3670 genomes were isolated, purified and sequenced previously [20]. The sequencing reads were trimmed using Trimmomatic (version 0.39) [21], and the trimmed reads were used to assemble genome contigs using SPAdes (version 3.15.4) [22]. The quality of the raw reads was compared with the quality of the trimmed reads and assembled reads using Quast (version 5.2.0) [23], quality of the assembled genome was also compared to a reference *S. maltophilia* genome of the strain NCTC10258 (Genbank accession number LS483377.1), Ragtag was used for automated assembly scaffolding using strain NCTC10258 as a reference [24]. Putative prophages within the genomes were detected and annotated using Phigaro (version 2.3.0) [25]

### 2.2. Pangenome Analysis

Sixty-seven complete *S. maltophilia* genomes with no irregularities in the genome were download from NCBI GenBank (accessed on 4 April 2024). The assembled sequenced genomes were annotated using Prokka (version 1.14.6) [26], pangenome analysis was done using Roary (version 3.11.2) [27]. Visualization of the pangenome analysis was done by Roary plots (version 0.1.0).

### 2.3. Anti-Phage Defense Systems and MGEs Analysis

Putative Horizontal Gene Transfer (HGT) events were predicted using Alien Hunter (version 1.7) [28], CRISPR/Cas Finder (version 4.2.20) was used to find CRISPR arrays and their associated Cas protein [29], the genomes were visualized using Proksee, anti-phage defense systems were found using defense-finder (version 1.2.2) [16] hosted on Galaxy platform [30]. Prophages were found using Phigaro (version 2.3.0) [25] MGEs were found using mobileOG-db [31] hosted on Proksee [32]. To estimate the degree of conservation of the anti-phage defense systems, the genes of each of the systems were aligned using MUSCLE alignment [33]; the resulting pairwise identity matrix was averaged to get a resulting average identity of each of the genes.

### 2.4. Determining Anti-Phage Defense Islands

A defense island was defined as the site flanked by five core genes up stream and five core genes downstream from the anti-phage defense systems and containing such systems in at least two *S. maltophilia* strains. The putative defense island should also contain MGE such as integration, excision or recombination related genes and/or prophages [17], the gene presence-absence comparison resulting from pangenome analysis was used as a starting place for determined core flanking genes surrounding each anti-phage defense system for each strain. The core flanking genes were compared with those present in the reference strain *S. maltophilia* NCTC10258. If the genes were present in the same order in both strains they were considered as the flanking genes of a defense island. Defense islands that contained only one defense system across all the strains were ignored.

## 3. Results

### 3.1. High Level of Heterogeneity of S. maltophilia Strains

A total of 1961 *S. maltophilia* genomes were downloaded from NCBI GenBank (https://www.ncbi.nlm.nih.gov/datasets/genome/?taxon=40324, accessed on 25 March 2024). Of them, 72 genomes were found to have complete assembly level with no atypical genomes and they were used for pangenome analysis. Among 72 *S. maltophilia* complete genomes, five genomes were obtained from strains isolated in Novosibirsk (*S. maltophilia* CEMTC2142, CEMTC2355, CEMTC3659, CEMTC3664, and CEMTC3670; all deposited in the Collection of Extremophilic Microorganisms and Type Cultures, CEMTC) that are hosts for two lytic phages: StM171 (20) and StenM174 (21). Pangenome analysis was performed for all the studied strains and for the locally isolated strains. Pangenome analysis of all the strains showed a high level of heterogeneity, with core genes shared between all the strains accounted for 2.1% of all genes, soft-core genes—2.2%, shell genes (shared by at least ten strains and at most 67 ones)—10%, and cloud genes (shared at most by ten strains) constituting the rest (Figure 1A,B). Independent genome analysis of five Novosibirsk strains expectedly revealed a much higher homogeneity; the percentage of core genes accounted for ~64% (Figure 1C). Data on the locally isolated strains and a detailed analysis of the number of genes shared between them are available (Appendix A), full comparison of the shared genes between the studied strains is available in Appendix A.

### 3.2. Defense Islands in the S. maltophilia Genomes

To determine the anti-phage systems-containing defense islands, previously established parameters [17] were used. The defense island was defined as a segment of the bacterial genome that contains an anti-phage defense system in at least two strains and is flanked by five core genes located both upstream and downstream. These core genes should appear in the same order in the strains that have the identified anti-phage system, and in the reference strain *S. maltophilia* NCTC10258.

The core flanking genes were mapped to the reference strain *S. maltophilia* NCTC10258 and 27 defense islands were identified (Figure 2A). These islands also contained prophages and genes related to recombination, transposition and integration. Notably, 23 defense islands were occupied in all studied strains; they could contain genes of various functions. For defense islands #5, #6, #20, and #22, empty instances were found in at least one *S. maltophilia*.

The frequency of occupancy of anti-phage system instances varied among different defense islands (Figure 2B). Four of the defense islands (#4, #13, #16, and #27) contained ~60% of the instances of the systems across the *S. maltophilia* strains. Thus, the defense island #27 was the only defense island occupied in the all the studied strains. It exhibited the highest number of anti-phage systems instances and was flanked by the *yfcA* and *mlaF* genes (Figure 3A). This island included instances of the RecBCD operon and Wadjet I system, which lacked the *jetD* gene. Defense island #27 accounted for 25% of all identified anti-phage defense system instances. Defense island #4, which was flanked by the genes *yceI* and *rcnR,* contained 13% of the instances of the *S. maltophilia* anti-phage defense systems. This island included most instances of RM type II system and instances of Azaca, Dazbog, Gao_Mza, dCTP terminase, and Menshen systems (Figure 3B). Defense island #16, bordered by the *rpoE* and *metR* genes, contained 12% of the instances of anti-phages systems, including most instances of Gabija and Pycsar systems and half of the instances of Septu, CBASS III, and RM type IIG system. This island consistently contained the integration-related gene *ligD* (Figure 3C). Finally, the defense island #13 accounted for approximately 9% of the instances of anti-phage defense systems. Flanked by the *guaA* and *pgrR* genes, this island was the most diverse in terms of the systems present (Figure 3D). Additional information about the flanking genes of the rest of the defense islands are shown in Appendix A.

### 3.3. Anti-Phage Defense Systems in S. maltophilia

The studied *S. maltophilia* genomes were searched for anti-phage defense systems, mobile genetic elements (MGE) and prophages. A total of 72 anti-phage defense systems and subsystems were found in the *S. maltophilia* genomes (Figure 4). These systems could be divided according to their functions: phage protein sensing systems (Avast, DSR, Gabija and Thoeris), abortive infection systems (AbiE, AbiD, AbiO, Lamassu-Fam, Dazbog and Retron), RM systems, RM-like systems (BREX and SspBCDE), toxin-antitoxin systems (DarTG, PD-Lambda, PD-T7, SanaTA, ShosTA, RosmerTA and Rst-PARIS), CRISPR-Cas systems, systems that rely on sequence independent detection of nucleotide acids (CBASS, Pycsar, recBCD and Wadjet), and other systems with unknown functions.

The defense systems varied in their frequency of occurrence; the most common system was RecBCD, which was present in all the studied strains. RecBCD is responsible for DNA repair and homologous recombination in bacteria; this operon also has an anti-phage function, as it expresses a nuclease that degrades both strands of the free DNA duplex into fragments of single-stranded DNA [34,35,36]. Another common system was Wadjet I found in 82% of the studied *S. maltophilia* strains. Wadjet I system is responsible for protective functions against transformation by circular plasmids and probably against infection by phage ssDNA [37]. The frequency of occurrence of this system in *S. maltophilia* is higher than its occurrence in bacteria that was estimated as 6% [38]. Wadjet I system was identified in two variants; the most frequent was probably inactive due to the absence of the *jetD* gene, the deletion of which leads to inactivation of the system [37]. The rest of the genes of the Wadjet I system (*jetA*, *jetB*, and *jetC*) are responsible for topology-based recognition of foreign DNA [37,38] and this incomplete Wadjet I version was conserved in defense island #27 along with the RecBCD operon.

Other anti-phage defense systems that were frequently found in *S. maltophilia* strains included RM type II (found in 49% of the studied strains), Gabija (45% strains, compared to 15% in bacteria in general [39]), RM type IV, type I and type IIG (found in 29%, 28%, and 24% of the studied strains, respectively), Septu, and the abortive infection system AbiE 4 (both were found in 20% of the strains) (Figure 4). Complete CRISPR-CAS system was found only in three local *S. maltophilia* genomes from Novosibirsk (*S. maltophilia* CEMTC 3659, 3664, and 3670); this system was type IIIA in the strains. The CRISRPR-CAS occurrence was substantially lower than that among bacteria (42%) [40] or in *Lacticaseibacillus rhamnosus* strains (around 45%) [41], but closer to that found in some species like *Bacillus cereus* (13.9%), which is known for its high adaptability to various environmental niches [42].

### 3.4. Variability of Defense Systems in S. maltophilia

To evaluate the degree of similarity of proteins in anti-phage defense systems across different strains, a pairwise alignment of the amino acids sequences between each two strains containing similar systems was performed. Then, the calculated values for this protein were averaged to obtain a single value that depicts how conservative the protein is across all the analyzed strains.

Among the most frequently found systems, RecBCD and AbiE 4 system’s proteins and proteins encoded by the genes *jetA*, *jetB*, and *jetC* of Wadjet I were found to be highly conservative (average MUSCLE amino acid pairwise alignment for each individual protein was >95%). Other frequently found systems, namely Gabija and all types of RM systems were found to be variable; average MUSCLE amino acid pairwise alignment values were ~50% for Gabija proteins, and <30% for proteins of RM system types I, II, IIG and IV. Among RM systems, only RM type III had a more conserved protein across the studied strains with pairwise alignment values ~70%.

Across the defense systems that were found in at least five strains, BREX I, Gao_MZA, and Shango were the only ones found to be highly conserved (average MUSCLE gene pairwise alignment was 98%, 97%, and 97%, respectively) (Figure 5). All the details regarding individual proteins for each system are available in Appendix A.

### 3.5. Defense Islands Occupation Pattern

To investigate the association between the phylogenetic relationship of *S. maltophilia* strains and the similarity of defense islands, the occupied islands were mapped to the strains on a phylogenetic tree. For 12 of the 27 defense islands, at least half of the studied strains with the same occupied island were phylogenetically related. The related strains usually shared the same anti-phage systems. So, defense island #9 was occupied in five strains, four of which were closely related phylogenetically and shared the same systems, namely RM type III, Rloc, and old_exonuclease. The fifth, unrelated strain had a different system (RST helicase) in this island. Similarly, defense island #19 was occupied by SoFic in phylogenetically related strains (Figure 6A) and defense island #20 contained BREX I and the complete variant of Wadjet I in four phylogenetically related strains (Figure 6B).

In contrast, most other defense islands were found to be occupied by the same systems in phylogenetically less related strains, which were isolated mostly from different locations. Examples include defense islands #6, #18, and #26, which are occupied by BREX I, Shedu, and Dsr I systems, respectively (Figure 7). Notably, *S. maltophilia* strains with similar defense islands #26 were isolated in the USA, China, and Kazakhstan (GCF_001274595.1, GCA_014076535, and GCA_024734725, respectively) (Figure 7B).

### 3.6. Variability of MGEs in Defense Islands

Various MGEs were found in defense islands in *S. maltophilia* strains. In most of the defense islands, genes encoding different types of transposases, integrase, gyrase, ligase and recombinase, which are responsible for the transfer of genomic islands between bacterial strains, were found. Prophages were identified in defense islands #2, #4, #6, #7, #12, #13, #20, #21, and #24, whereas putative phage satellites were detected in defense islands #6, #7, #8, #12, #16, #20, and #24 (Figure 8). Notably, the *ihfB* gene, part of the integration host factor complex [43], was conserved flanking defense island #13. tra and trp operons, which are important for plasmid transmission, were found in defense islands #12, #13, #16, #18, #19, #22, and #25. Finally, it was revealed that the tRNA genes, which may represent a possible integration site, were conserved within or flanking most defense islands, with the exception of defense islands #1, #2, #8, #14, and #15 (Figure 8). Defense islands #5, #15, #17, and #23 had no elements linked to MGEs and no tRNA genes flanking or within them.

## 4. Discussion

The rise of antibiotic resistance and the increased importance of potential use of phages in therapy make understanding bacterial anti-phage defense mechanisms relevant for basic research and for clinical settings. Previous studies have shown that using different lytic phages against the same bacterial strains could have varying outcomes, with some phages demonstrating weak lytic activities and burst size [44]. This can make the phages less effective in therapy. The poor lytic activity of the phages may be explained by the features of the phage itself, and it could also be linked to the various anti-phage defense systems that are present in the targeted bacterial strains [45]. So, detecting the most common defense systems of a bacterial species and their possible genetic locations provides several advantages: first, phages that are chosen for a cocktail against a target infectious agent could be selected on the basis of their ability to counter the most frequently found defense mechanisms in the bacterial species. Thus, RM systems of various types are common in *S. maltophilia* complex, and phages that have 7-deazaguanine modification pathway [46] would be favorable for therapeutic cocktails. Identifying the defense islands where anti-phage systems tend to aggregate allows for discovering novel ones in the future.

As *S. maltophilia* was only recently identified as a pathogenic infectious agent for immunocompromised patients, its anti-phage defense systems have not been previously studied in details. Here, we provide a thorough examination of the anti-phage systems found in *S. maltophilia* strains. Despite the high heterogeneity of *S. maltophilia* (only 4.3% of the genes are core and soft-core genes, compared to 13% in *E. coli* [47] and 15% in *P. aeruginosa* [48]), most of the anti-phage systems were found in four of 27 identified defense islands. So, the frequency of occupancy of anti-phage system instances in *S. maltophilia* differs from those revealed in *E. coli,* where the occupation rate of the 41 defense islands was mostly equal [17] and *P. aeruginosa* where the defense systems were found only in two defense cores [18].

When studying preservation of anti-phage defense systems of *S. maltophilia* and their defense islands, it was determined that the encoded proteins varied widely between the strains for most systems. This fact can be explained by the adaptation of *S. maltophilia* to diverse phages. The presence of variability in the defense genes suggests that the bacteria are under permanent pressure to adapt and diversify their anti-phage mechanisms to keep up with phage evolution. New variants of defense systems can emerge due to co-evaluation of phages and their bacterial hosts; so, the bacteria acquire mutations in response to phage counter-defenses. Strain-specific variations in the defense genes may correlate with the phage types encountered by each strain; probably, bacteria accumulate mutations that provide resistance against the predominant phages in their local environment. Conserved systems were rare and were limited by BREX I, RecBCD, Abie 4, and part of Wadjet I system that is responsible for detecting foreign DNA by its topology. Some anti-phage systems found in *S. maltophilia* may be inactive.

In many cases, identical anti-phage systems were found in the same defense islands across *S. maltophilia* strains that are neither taxonomically related, nor isolated in the same geographical location. The possible explanation for this fact is that these defense systems were inherited from an ancestral strain, and were preserved due to their advantage to the strain. Anti-phage systems of *S. maltophilia* were rarely found to be shared among strains based on their isolation site if they were not phylogenetically related. These data possibly suggest that the drivers for the emergence or acquiring of those systems are not the immediate presence of certain phages in the environment, from which the strains were isolated, but rather a long evolutionary process that is less dynamics and more linked to accumulating relevant systems. The tendency to find the same anti-phage defense systems accumulated in the same defense islands requires further investigation.

The identification of anti-phage systems does not mean that the studied bacterial strain is completely immune to phages intended for possible therapeutic use. These defense systems may not function properly or may only reduce the infectious ability of the phage, rather than eliminate it. Thus, despite the existence of a wide range of anti-phage defense systems in locally isolated *S. maltophilia* strains (Figure 9), two phages that were studied previously showed lytic activity, albeit varying. One of the phages, StM171, had weak lytic activity [20], whereas the other, StenM174, was highly lytic and, therefore suitable for therapy, despite the presence of these defense systems [49].

Finally, only 72 complete genomes of *S. maltophilia* were involved in this study. When more complete *S. maltophilia* genomes are available, some additional elements of anti-phage defense systems can be found and some new tendency can be determined.

## Figures and Tables

**Figure 1 viruses-16-01903-f001:**
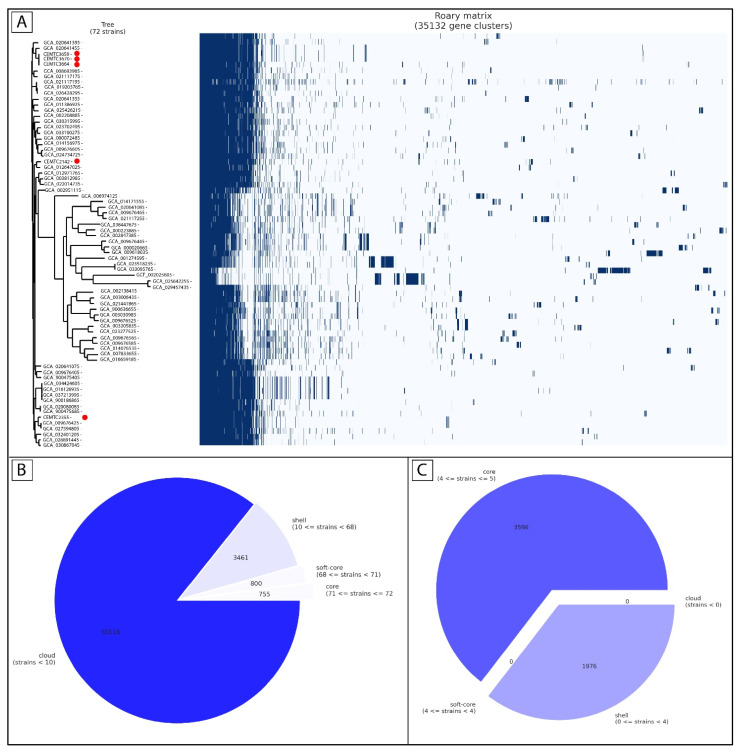
Pangenome analysis of *S. maltophilia* strains. (**A**) Matrix of the shared genes between 72 studied strains with taxonomy relatedness; local strains are marked with red circles. (**B**) Distribution of core and accessory genes in 72 *S. maltophilia* strains. (**C**) Distribution of core and accessory genes in five *S. maltophilia* strains from Novosibirsk.

**Figure 2 viruses-16-01903-f002:**
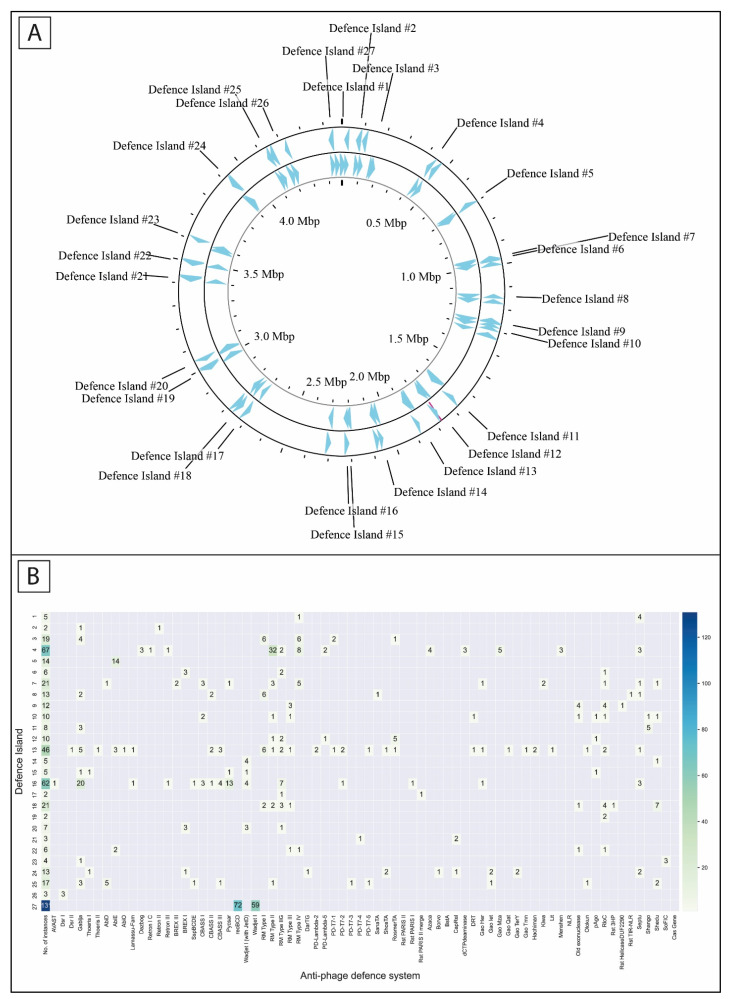
Putative defense islands of anti-phage defense systems in *S. maltophilia*. (**A**) The defense islands mapped to *S. maltophilia* reference strain NCTC10258. (**B**) The distribution of anti-phage systems in defense islands.

**Figure 3 viruses-16-01903-f003:**
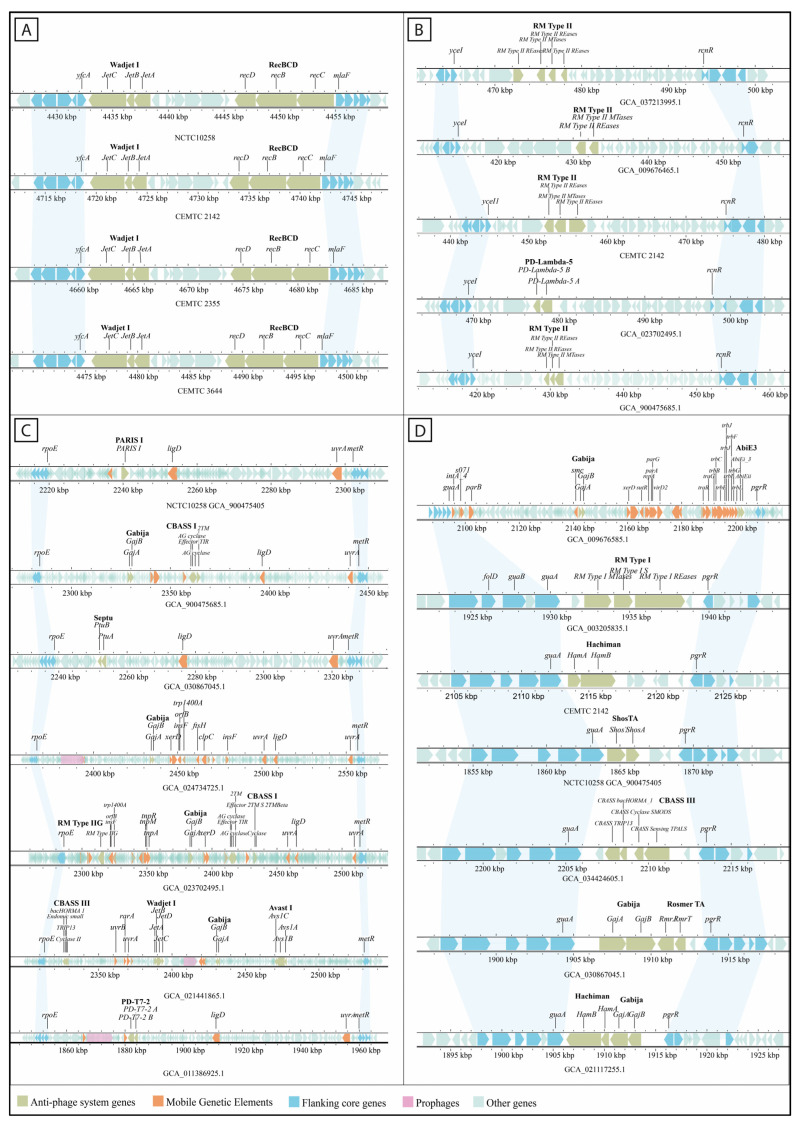
Defense islands with the highest frequency of occupancy. (**A**) Defense island #27, occupied by Wadjet I and RecBCD systems. (**B**) Defense island #4 containing most of RM type II instances and other anti-phage systems. (**C**) Defense island #16 containing a wide range of various anti-phage defense systems. (**D**) Defense island #13 containing the most diverse set of anti-phage systems across the studied *S. maltophilia* strains.

**Figure 4 viruses-16-01903-f004:**
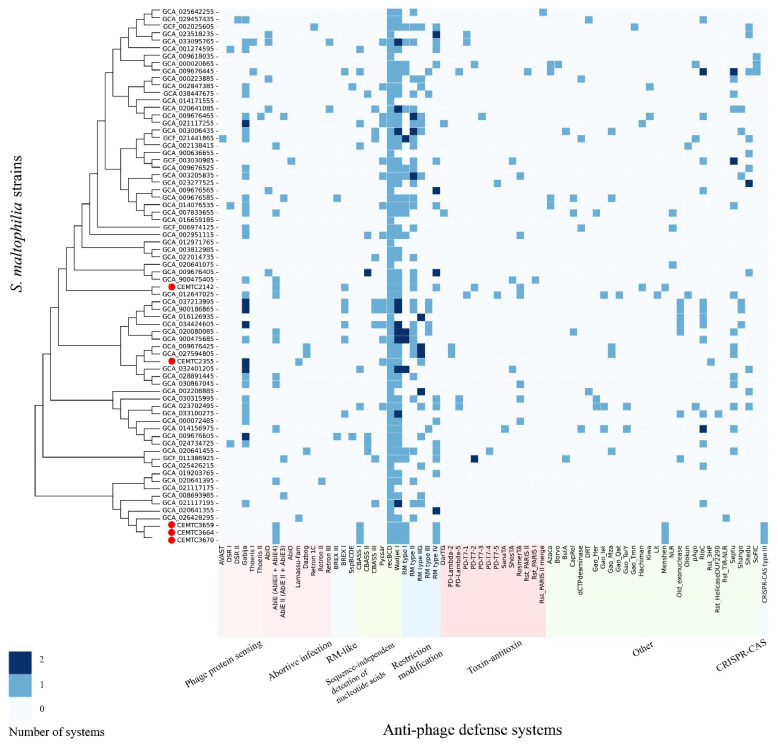
Instances of defense systems found in *S. maltophilia* strains. Local genomes are marked with red circles.

**Figure 5 viruses-16-01903-f005:**
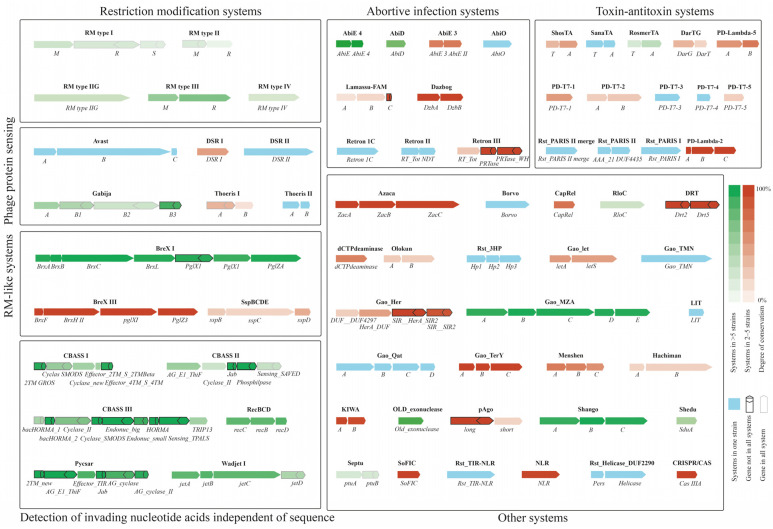
Classification and conservation of anti-phage defense systems found in *S. maltophilia*. Blue color marks systems present in one strain, red—in 2–5 strains, green—systems present in more than 5 strains. The genes with bolded borders refer to genes that were not present in all the instances of their systems.

**Figure 6 viruses-16-01903-f006:**
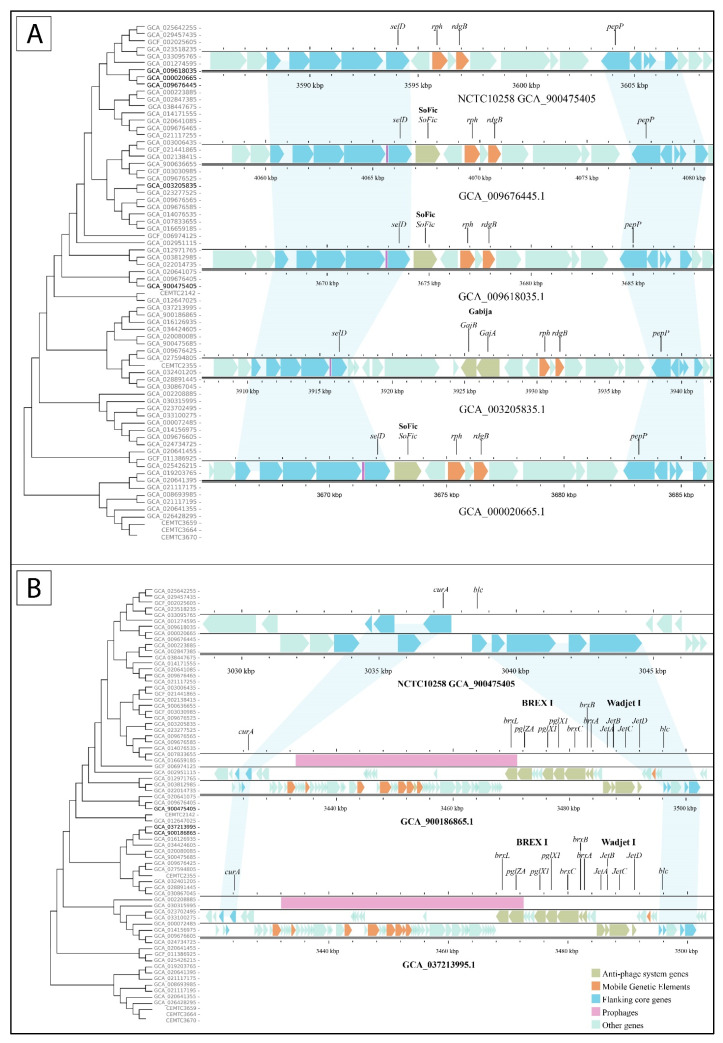
Defense islands occupied by the same anti-phage systems in phylogenetically related strains. (**A**) Defense islands #19 sharing SoFic system across four relative strains. (**B**) Defense island #20 containing identical anti-phage systems in two phylogenetically related strains. Reference strain NCTC10258 with empty defense island #20 is shown above.

**Figure 7 viruses-16-01903-f007:**
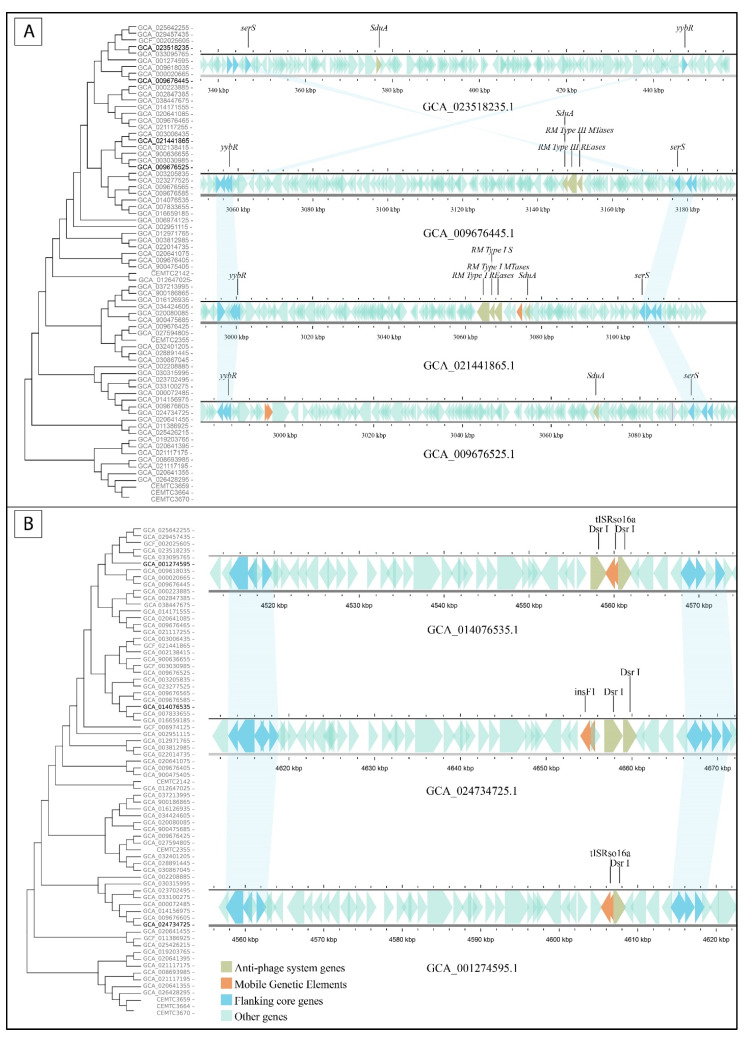
Examples of defense islands of phylogenetically unrelated strains that are occupied by the same anti-phage defense systems: (**A**) defense island #18 is occupied by Shedu, (**B**) defense island #26 is occupied by Dsr I.

**Figure 8 viruses-16-01903-f008:**
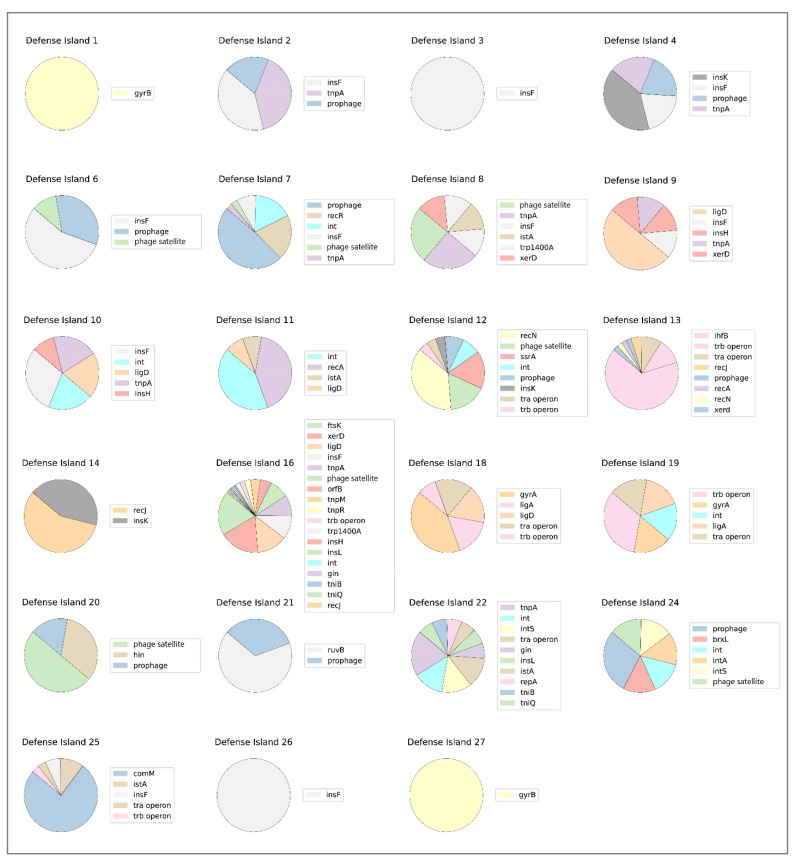
Genetic elements linked to the movement of the defense islands between the bacterial strains. Defense islands #5, #15, #17, and #23 are not shown as they do not contain elements associated with carrying defense systems between bacterial genomes.

**Figure 9 viruses-16-01903-f009:**
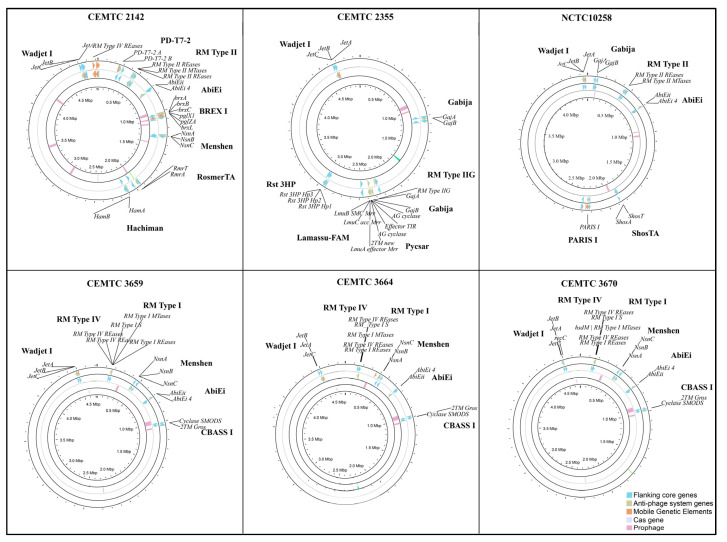
Anti-phage defense systems (outer rings of each circle), prophages (middle ring) and CAS genes (inner ring) in the *S. maltophilia* genomes of reference strain NCTC10258 and local strains from Novosibirsk (CEMTC 2142, CEMTC 2355, CEMTC 3659, CEMTC 3664, CEMTC 3670).

## Data Availability

All data related to the research results are provided in the Appendix A.

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
