# Peer review of "Genome Analysis of Anti-Phage Defense Systems and Defense Islands in *Stenotrophomonas maltophilia*: Preservation and Variability"

_viruses, 2024, doi:10.3390/v16121903_

Round 1

Reviewer 1 Report

Comments and Suggestions for Authors

The manuscript ‘Genome Analysis of Anti-Phage Defense Systems and Defense Islands in Stenotrophomonas maltophilia: Preservation and Variability’ presents a comprehensive pangenome analysis of 72 complete Stenotrophomonas maltophilia genomes obtained from NCBI GenBank, including five newly sequenced strains isolated from Western Siberia. The authors focused on identifying the anti-phage defense systems present in S. maltophilia, examining their genomic localization, distribution, and conservation. As a result, 27 defense islands were identified and thoroughly examined.

This research provides valuable insights into the defense mechanisms of S. maltophilia, facilitating the rational selection of candidate bacteriophages for the development of phage therapy against this pathogen.

Overall, the manuscript is clearly written and understandable.

Minor issues/suggestions

1.       A recent publication on the same research topic has been released: Zhuang et al., Genomic insights into the phage-defense systems of Stenotrophomonas maltophilia clinical isolates, Microbiological Research, Volume 278, 2024. Although the analysis of S. maltophilia anti-phage systems, presented by Zhuang et al., is less comprehensive, the reviewer believes that the authors should discuss this publication.

2.       The names of genes in Figures 3 and 5 are currently unreadable due to the small font size and low resolution.

3.       The authors should consider adding a brief description of the anti-phage systems analyzed to the suitable sections of the manuscript or citing the appropriate source publications.

Page 3, line 122:

An incorrect publication (reference 21) has been cited. Reference 21 does not discuss phage StenM174. It seems that reference 49 should be mentioned here.

Page 4, Figure 1B

The black digits on the dark blue background are nearly invisible. Please consider changing the color to improve the readability of Figure 1B.

Page 6, Figure 2B

The authors should consider expanding the legend for Figure 2B to ensure that understanding the results presented in the figure does not require referring to the manuscript (section 3.2).

Page 11, Figure 4A and legend

“(A) Defense islands #19 sharing SoFic system across four relative strains.” – it seems that only three relative strains sharing SoFic system are presented in Figure 6A.

Author Response

Reply to the Reviewer 1:

We would like to thank you for the positive review.

Comment 1:  A recent publication on the same research topic has been released: Zhuang et al., Genomic insights into the phage-defense systems of Stenotrophomonas maltophilia clinical isolates, Microbiological Research, Volume 278, 2024. Although the analysis of S. maltophilia anti-phage systems, presented by Zhuang et al., is less comprehensive, the reviewer believes that the authors should discuss this publication.

Response 1: Indeed, the above article describes the first insight into the S. maltophilia anti-phage systems. We add this reference and data from the article in the Results section when discussing the most prevalent anti-phage defense systems:” The distribution of Gabija, Abi and RM systems was close to that found in recent study of clinical S. maltophilia strains isolated in China [42] (45%, 20%, 90%, respectively). Line 226. In addition, we discussed this article in the Discussion section.

Comment 2: The names of genes in Figures 3 and 5 are currently unreadable due to the small font size and low resolution.

Response 2: The font size was increased.

Comment 3: The authors should consider adding a brief description of the anti-phage systems analyzed to the suitable sections of the manuscript or citing the appropriate source publications.

Response 3: Due to the relatively large number of anti-phage defense systems included in the paper, we thought that adding description for them will take a large part of the introduction, thus, we decided to describe them in terms of their general classification that is based on their functions.

Comment 4: Page 3, line 122:

An incorrect publication (reference 21) has been cited. Reference 21 does not discuss phage StenM174. It seems that reference 49 should be mentioned here.

Response 4: citation was corrected

Comment 5: Page 4, Figure 1B

The black digits on the dark blue background are nearly invisible. Please consider changing the color to improve the readability of Figure 1B.

Response 5: font size was increased and the colors were modified.

Comment 6: Page 6, Figure 2B

The authors should consider expanding the legend for Figure 2B to ensure that understanding the results presented in the figure does not require referring to the manuscript (section 3.2).

Response 6: the following explanation was added to the legends: most defense systems were present more than once across the studied strains, the numbers in the chart refer to the instances found in each corresponding defense island. Additionally, axis labels font sizes were increased

Comment 7: Page 11, Figure 4A and legend

“(A) Defense islands #19 sharing SoFic system across four relative strains.” – it seems that only three relative strains sharing SoFic system are presented in Figure 6A.

Response 7: the legend was corrected to: “sharing SoFic system across three related strains”.

Reviewer 2 Report

Comments and Suggestions for Authors

The manuscript entitled "Genome analysis of anti-phage defense system and defense island in Stenotrophomona maltopia: preservation and variability" by Jdeed et al., provides a comprehensive result of the analysis of the anti-phage systems repertoire and the defense islands in the Stenotrophomona maltopia. The research question of the article is clear, the methodology is sound, and the findings are well-articulated and supported by evidence. Overall, I found the manuscript to be well-written, engaging, and insightful. They described high level of heterogeneity in the anti-phage defense system of Stenotrophomona maltopia. It is very interested that they included into analysis also five recently strains isolated in Novosibirsk. Evan if they did not find any obvious association between the pattern of distribution of the anti-phage defense systems and phylogenetic features, this article provides very useful information about anti phage-defense systems. The manuscript tackles a highly relevant issue, offering novel insights that are significant for researchers and practitioners in the field. I believe this article will serve as a valuable guide for future discoveries of anti-phage defense systems. For researcher in the field, this article will be very helpful.

My suggestion: For readers who are not familiar with S maltiophilia, it would be interesting to explain in the introduction what symptoms it causes in immunocompromised patients.

Author Response

Reply to the Reviewer 2:

We are grateful for the positive evaluation of our manuscript.

Comment 1: My suggestion: For readers who are not familiar with S maltiophilia, it would be interesting to explain in the introduction what symptoms it causes in immunocompromised patients.

Response1: The following sentences were added “S. maltophilia mainly causes nosocomial infections in immunocompromised patients with weakened immune systems, namely infections of the bloodstream, respiratory system, and urinary tract [6]. This bacterium can form biofilms in tubes, catheters and other medical devices used to treat patients, as well as on the surface of prosthetic materials. Line 32.

Reviewer 3 Report

Comments and Suggestions for Authors

     The paper presents an overview of the antiphage defense systems of Stenotrophomonas maltophilia based on 72 available complete genome sequences of this bacterium. The authors performed a pan-genome analysis of 72 complete genomes and found that the core genome accounts for 2.1% of all genes. The authors then analyzed the genome of the type strain for the presence of a defense island - " a segment of the bacterial genome that contains an anti-phage defense system in at least two strains and is flanked by five core genes located both upstream and downstream." In the type strain of S. maltophilia, 27 such islands were found. In the analyzed complete genomes of the S. maltophilia species, the authors found 72 different antiphage defense systems within the identified islands. The authors then analyzed the organization of the islands and variations in the occurrence of antiphage defense systems in them.

Major comments:

(1) The paper is illustrated with 9 figures. In the context of the paper, all the figures are informative, but the text in figures 3, 5 and 9 is not readable and it is impossible to see the details in these figures. As a reviewer, I only have access to the pdf version. I recommend increasing the resolution by a factor of two for figures 3 and 9 (i.e. if the resolution was 600 dpi, please change it to 1200 dpi). For figure 5, the subfigures can be rearranged to fill the entire page to increase the size of the details.

(2) In January 2024, Zhuang et al. published a manuscript, “Genomic insights into the phage-defense systems of Stenotrophomonas maltophilia clinical isolates”. What novelty does this manuscript contain compared to a previously published article?

Minor comment:

It will be good for the readers to identify that the strain NCTC10258 is type strain of S. maltophilia species in material and methods section.

Author Response

Reply to the Reviewer 3:

We are grateful for your advises and used all of them to improve our manuscript.

Comment 1: The paper is illustrated with 9 figures. In the context of the paper, all the figures are informative, but the text in figures 3, 5 and 9 is not readable and it is impossible to see the details in these figures. As a reviewer, I only have access to the pdf version. I recommend increasing the resolution by a factor of two for figures 3 and 9 (i.e. if the resolution was 600 dpi, please change it to 1200 dpi). For figure 5, the subfigures can be rearranged to fill the entire page to increase the size of the details.

Response 1: Agree. So, the font sizes were increased and the figures were modified.

Comment 2: In January 2024, Zhuang et al. published a manuscript, “Genomic insights into the phage-defense systems of Stenotrophomonas maltophilia clinical isolates”. What novelty does this manuscript contain compared to a previously published article?

Response 2: While the paper published by Zhuang et al. includes an overview of the anti-phage defense systems found in the locally isolated strains they studied and mentions some of the prevalent ones, it places greater emphasis on prophage analysis and pangenome analysis, focusing primarily on clinical strains that are locally isolated. We would like to discuss several points that distinguish the focus of our paper:

  1. Our selection of studied strains is more widespread; in addition to analyzing five locally isolated and sequenced strains, we examined 67 strains from GenBank with complete genomes that were isolated from geographically unrelated locations.
  2. This broader selection allowed us to draw more generalized conclusions about the consistency of accumulation of anti-phage defense systems at specific sites within the bacterial genomes. Consequently, we studied the defense islands of S. maltophilia and the mobile genetic elements that facilitate their transfer between strains in greater detail.
  3. We also investigated the presence and distribution of defense islands and explored the phylogenetic relationships among strains isolated from diverse locations. Our findings highlight the likelihood of similar systems occupying the same defense islands across different geographic regions.
  4. Additionally, we analysed the preservation of each protein that constitutes these defense islands, and provided insights into their adaptability.

We add this reference and data from the article in the Results section when discussing the most prevalent anti-phage defense systems:” The distribution of Gabija, Abi and RM systems was close to that found in recent study of clinical S. maltophilia strains isolated in China [41] (45%, 20%, 90%, respectively). Line 226. In addition, we discussed this article in the Discussion section.

This focus on defense systems enables us to draw meaningful conclusions about the consistency of defense island sites and their varying preservation. Ultimately, we believe our paper offers valuable insights that facilitate more rational selection of phages for therapy, emphasizing the importance of understanding anti-phage defense mechanisms in developing effective phage treatment strategies.

Comment 3: It will be good for the readers to identify that the strain NCTC10258 is type strain of S. maltophilia species in material and methods section.

Response 3: Agree. However, this strain, NCTC10258, was identified as the reference strain for S. maltophilia throughout the text. In the corrected version, we added for clarity: “..scaffolding using the type strain NCTC10258 as a reference”. in lines 82 and 84 in the Materials and Methods section